# Synthesis, Biophysical Interaction of DNA/BSA, Equilibrium and Stopped-Flow Kinetic Studies, and Biological Evaluation of bis(2-Picolyl)amine-Based Nickel(II) Complex

**DOI:** 10.3390/biomimetics7040172

**Published:** 2022-10-22

**Authors:** Esraa Ramzy, Mohamed M. Ibrahim, Ibrahim M. El-Mehasseb, Abd El-Motaleb M. Ramadan, Fawzia I. Elshami, Shaban Y. Shaban, Rudi van Eldik

**Affiliations:** 1Chemistry Department, Faculty of Science, Kafrelsheikh University, Kafrelsheikh 33516, Egypt; 2Chemistry Department, Faculty of Science, Taif University, P.O. Box 11099, Taif 21944, Saudi Arabia; 3Department of Chemistry and Pharmacy, University of Erlangen-Nuremberg, 91058 Erlangen, Germany; 4Faculty of Chemistry, Nicolaus Copernicus University in Torun, 87-100 Torun, Poland

**Keywords:** DNA/protein affinity, association/dissociation, metallodrugs, nickel complexes, biological activity

## Abstract

Reaction of bis(2-picolyl)amine (**BPA**) with Ni(II) salt yielded [(BPA)NiCl_2_(H_2_O)] (**NiBPA**). The Ni(II) in **NiBPA** bound to a **BPA** ligand, two chloride, and one aqua ligands. Because most medications inhibit biological processes by binding to a specific protein, the stopped-flow technique was used to investigate DNA/protein binding in-vitro, and a mechanism was proposed. **NiBPA** binds to DNA/protein more strongly than **BPA** via a static quenching mechanism. Using the stopped-flow technique, a mechanism was proposed. BSA interacts with **BPA** via a fast reversible step followed by a slow irreversible step, whereas **NiBPA** interacts via two reversible steps. DNA, on the other hand, binds to BPA and **NiBPA** via the same mechanism through two reversible steps. Although BSA interacts with **NiBPA** much faster, **NiBPA** has a much higher affinity for DNA (2077 M) than BSA (30.3 M). Compared to **NiBPA**, BPA was found to form a more stable BSA complex. When **BPA** and **NiBPA** bind to DNA, the Ni(II) center was found to influence the rate but not the mechanism, whereas, for BSA, the Ni(II) center was found to change both the mechanism and the rate. Additionally, **NiBPA** exhibited significant cytotoxicity and antibacterial activity, which is consistent with the binding constants but not the kinetic stability. This shows that in our situation, biological activity is significantly more influenced by binding constants than by kinetic stability. Due to its selectivity and cytotoxic activity, complex **NiBPA** is anticipated to be used in medicine.

## 1. Introduction

Metal complexes with the ability to bind and react with specific DNA locations under physiological conditions are of great interest in the development of anticancer drugs [1]. By attaching to and cleaving DNA, metal complexes inhibit tumor growth [2]. Manufactured nucleases have lower molecular weights and are less reactive under certain conditions than natural DNA-binding enzymes [3,4]. Furthermore, certain transition metal compounds are being studied for therapeutic purposes [5,6]. Nickel complexes have been found to play an important role in the field of metal-based chemotherapeutics due to their biocompatibility [7]. Nickel is also an essential component of life, as it can be found in a variety of enzymes [8]. Nickel (II) complexes, on the other hand, are thought to be one of the most promising anticancer drug alternatives to cisplatin [9,10,11]. The precise mechanism of anticancer activity, on the other hand, is still unknown. In general, clinically approved anticancer treatments are chemicals that either directly damage DNA or indirectly block DNA synthesis by inhibiting nucleic acid precursor production. They have the potential to interfere with the hormonal stimulation of cell growth [12,13]. As a result, considerable efforts are being extended to investigate the interaction of nickel(II) complexes with DNA and their cytotoxic effects [14,15,16,17,18]. Although research has been conducted to better understand the interactions of metal complexes, metal ions, and Schiff base ligands with albumins, it appears that research on nickel(II) Schiff base complexes is limited [19,20,21].

Polypyridyl ligands are very interesting in biological and materials science [22,23,24,25,26,27,28,29]. Tridentate ligands with two identical terminal coordination sites, such as bis(2-picolyl)amine (BPA), are being studied as building blocks in a variety of areas, including chemosensors, catalysts, and biomedicinal applications [30,31,32,33,34,35]. BPA and its secondary nitrogen-substituted derivatives develop stronger complexes with nickel(II), showing the coordination versatility of such ligands and their significance in a variety of biological, catalytic, and magnetic aspects [36,37,38,39,40]. Numerous authors have also explored the use of Ni(II) complexes based on BPA for the cleavage of DNA or RNA [14]. On the other hand, drug interactions at the protein binding level have a significant impact on the apparent distribution volume and elimination rate. As a result, the interactions of metal complexes with serum albumins have received a great deal of attention in the scientific community, with researchers focusing on antitumor, metallopharmaceutical, pharmacokinetics, and structure–activity relationships [41].

BSA is the most extensively researched serum albumin due to its structural similarity to human serum albumin (HSA). This finding emphasizes the importance of studying the interaction behavior of metal complexes with BSA while evaluating their anticancer properties. Our journey with detailed mechanistic studies on DNA/protein binding recently began with the challenge of exploring this type of interaction via related systems while also attempting to answer the question, is the metal center important for binding? [42,43,44,45]. It was discovered that a metal center not only increases ligand affinity to DNA/protein but also alters the binding mechanism. When a ligand and its Ti(IV) complex were mixed with chitosan nanoparticles, the Ti(IV) center was found to slow the affinity but not change the mechanism. These findings suggest that the metal center is not always necessary for binding. 

Based on these intriguing findings, and because of the role and activity of nickel and its complexes in biological systems [46], as well as the importance of bis(2-picolyl)amine (**BPA**) in medicine, we describe the synthesis and characterization of [(BPA)NiCl_2_(H_2_O)] (**NiBPA**). Furthermore, a detailed kinetic investigation of the interaction of **BPA** and its Ni(II) complex (**NiBPA**) with ct-DNA/protein has been performed using UV/vis, fluorescence, and the stopped-flow technique, and a mechanism was reported. To evaluate the antibacterial activities of both compounds, *Escherichia coli*, *Salmonella choleraesuis*, *Staphylococcus aureus,* and *Proteus mirabilis* bacteria strains were chosen. The minimum inhibitory concentration (MIC) and minimum bactericidal concentration (MBC) of the compounds against these bacteria were tested in vitro. Additionally, the cytotoxicity against HepG2 human liver cancer cells is reported, and the results are correlated with the binding constants. 

## 2. Experimental

### 2.1. Materials and General Methods

All of the chemicals used were of the highest analytical grade. Merck Chemical Company provided the ligand **BPA**, which was used without further purification. The IR spectra were captured using a Bruker Alpha-Atunated FT-IR Spectrophotometer in the 400–4000 cm^−1^ range. Shimadzu spectrophotometer UV-240 was used to obtain electronic absorption spectra in a methanolic solution. At room temperature, magnetic moments were measured using Gouy’s method.

### 2.2. Synthesis of the NiBPA

At 60 °C, a solution of Ni(II)Cl_2_ (0.13 g, 1.0 mmol) in ethanol (20 mL) was heated, and a solution of BPA (0.2 g, 1.0 mmol) in ethanol (20 mL) was added. The mixture was heated for two hours. After cooling to room temperature, 40 mL of n-hexane was added, and the resulting precipitate was isolated by filtering to give **NiBPA**: Yield: 75% (0.14 g, 0.4 mmol). Elemental Analysis calculated for C_12_H_15_Cl_2_N_3_NiO; C, 41.55; H, 4.36; N, 12.11; Found: C, 41.22; H, 4.10; N, 12.03. IR (KBr, cm^−1^): ν(NH): 3164, 1517. ν(C=N)_met_: 1678, ν(M-N=C): 721, ν(M–N): 492. *Λ*_m_ = 20 Ω^−1^mol^−1^cm^2^.

### 2.3. DNA Interaction Studies

Electronic absorption measurements were performed with a constant concentration of ctDNA and increasing concentrations of **BPA** and **NiBPA** (Na-phosphate buffer solution, pH 7.2). **BPA** and **NiBPA** were dissolved in DMSO and diluted in phosphate buffer (pH 7.2) (DMSO percent by v/v does not exceed 3%). Intrinsic fluorescence was measured with the ct-DNA solution at the excitation wavelength (ex, 330), and emission spectra were recorded at 550–700 nm after aliquot addition of BPA and **NiBPA**.

### 2.4. Protein Binding Studies

The excitation and emission wavelengths of BSA at 280 nm were monitored in protein binding studies using BSA (0.5 M) solution prepared in Tris-HCl buffer (pH 7.5), and the solution was stored in a dark place at 4 °C.

### 2.5. Protein and ctDNA Kinetic Investigation Studies

The kinetics of protein and DNA binding was measured using an Applied KinetAsyst SF-61DX2 stopped-flow instrument. At 25 °C, equal volumes of compounds against protein and DNA solutions were rapidly mixed in the stopped-flow instrument. The binding reactions were studied using a 10-fold excess of the ligand **BPA** and **NiBPA** under pseudo-first-order conditions. For each experimental condition, all recorded rate constants were taken into account, from which the mean values of at least three independent kinetic runs were calculated.

### 2.6. Cell Viability Assay

Methyl thiazolyl tetrazolium (MTT) colorimetry [47] was used to assess the cytotoxicity of **BPA** and one for human liver cancer (HepG2) cell lines. MTT can be converted into formazan by living cells with an active metabolism, but this ability is lost in dying cells. Therefore, in this study, the MTT assay was used to determine how many viable cells there were. Before the investigated compounds were added, cells were cultured in 96-well flat-bottomed microtiter plates at a density of 1 × 10^4^ cells/well (100 l/well) for 70 to 80 percent confluent cultures, and the cultures were then incubated at 37 °C and 5 percent CO_2_ for 24 h. The compounds were used in a range of concentrations (31.2, 62.5, 125.0, 250.0, 500.0, and 1000.0 g/mL), and the cells were cultured for 24 h. Then, 10 l of the 12 mM MTT stock solution (5 mg/mL MTT in sterile PBS) was added to each well at the conclusion of the incubation period. After that, the plate was incubated at 37 °C for 4 h. The purple formazan crystal that formed at the bottom of the wells was removed, and 100 l of DMSO was used to dissolve it for 20 min. There was also a negative control that involved adding 10 mL of the MTT stock solution to 100 mL of medium on its own. On an ELISA reader, the absorbance at 570 nm was read (StatFax-2100, Awareness Technology, Inc., USA). The percentage of cells inhibited was calculated using:$$\%\;\mathrm{Cell}\mathrm{Inhibition}=100-\frac{T-T_{0}}{C-T_{0}}$$
where *T* is the optical density of the test sample, *T*_o_ is the optical density at time zero, and C is the optical density of the control.

## 3. Results and Discussion

### 3.1. Synthesis and Structural Characterization

The neutral complex [(BPA)NiCl_2_(H_2_O)] (**NiBPA**) was produced in good yield by treating hydrated NiCl_2_ with one equivalent of BPA in boiling ethanol at 60 °C. IR and UV-vis spectroscopies, as well as elemental analysis, were used to characterize complex **NiBPA**. Figure 1 depicts the electronic spectrum of **NiBPA**, which shows an intense peak at λ_max_ = 284 nm, indicating the inter ligand π→π* transition in the **BPA** ligand. A less intense band at 477 nm, on the other hand, is attributed to the nickel d_π__→__π_* MLCT transition. At 688 nm, a significant low-energy absorption band assigned for the d–d transition was seen. This may be assigned for the ^3^A_2g_(F) to ^3^T_1g_(F)(v2) transition and support the octahedral geometry of the nickel complex [19]. The band is observed at around 1603 cm^−1^ in the IR spectra of **NiBPA** (Figure 1), corresponding to vibrations of the C=N bond in the pyridine ring, shifted to higher frequencies, compared with free **BPA**, ν(C=N) = 1583 cm^−1^, and indicates coordination of nickel (II) to the **BPA** ligand. The vibration bands at 3050 and 1471 cm^−1^, ascribed for the ν(N–H) of free **BPA**, shifted to a higher frequency and appeared at 3238 cm^−1^ and 1485 cm^−1^ for **NiBPA**, respectively. A broad band at 3470 cm^−1^ was assigned to the coordinated water stretching vibration. Conductivity measurements of **NiBPA** were carried out in MeOH and DMF solution, and the *Λ*_m_ values are in the range of 22 and 18 Ω^−1^ mol^−1^ cm^2^, respectively. These values indicate that **NiBPA** is neutral with the participation of two chloride anions in the coordination sphere. The magnetic moment of **NiBPA** was found to be 2.87 B.M, confirming its paramagnetic character with two unpaired electrons and further confirming the octahedral configuration [48]. 

**NiBPA** proved to be very difficult to obtain suitable single crystals. A number of structure determinations attempted always resulted in high residual electron density maxima close to the central nickel atoms and showed the presence of numerous solvent molecules. However, the overall connectivity and geometry of **NiBPA** could be established (Figure 2). A preliminary structure of **NiBPA** reveals that the single Ni(II) center is tridentate bound to the **BPA** ligand, with three additional coordination sites occupied by terminally bound two chloride and one aqua ligands. One chloride is *trans* to the secondary amine, and the other is trans to one of the pyridine moieties. Oxygen from water is *trans* to the other pyridine moiety, and the nickel center has an approximately octahedral geometry. 

### 3.2. Protein Binding Studies

Interactions between the most common blood protein, serum albumin, and complexes have sparked a lot of interest recently [49] because of their structural similarity to human serum albumin. To better understand the mechanism of interaction between **BPA** and its nickel(II) complex **NiBPA** with BSA, fluorescence quenching studies were carried out. BSA’s fluorescence is due to intrinsic protein characteristics, such as the presence of tryptophan and tyrosine residues. Variations in the emission spectra caused by protein conformational changes, subunit interaction, substrate binding, or denaturation are largely attributed to the tryptophan residue [50,51]. As a result, BSA fluorescence behavior can provide valuable insight into protein structure, dynamics, and folding. Fluorescence titration experiments were performed at room temperature with BSA and different concentrations of **BPA** and **NiBPA** in the range 200–600 nm (λ_ex._ 280 nm). A slight red shift (40 percent, BPA; 65 percent, **NiBPA**; Figure 3) reduced the intensity of BSA fluorescence at 343 nm. The presence of the protein’s active site in a hydrophobic environment causes the redshift. It was suggested that **BPA** or **NiBPA** and the BSA protein have some sort of interaction [50,51]. The quenching mechanism can be set to either static or dynamic operation. Dynamic quenching occurs when the fluorophore and the quencher come into contact during the excited state’s transient existence, whereas static quenching is caused by the formation of a complex between the quencher and the fluorophore in the ground state. BSA (fluorophore) has a UV-vis absorption spectrum that can be used to investigate the type of quenching quickly and easily. Dynamic quenching does not cause a significant change in the fluorophore’s absorption spectra; however, static quenching does cause fluorophore perturbation [49,50,51]. Figure 4 shows that the intensity of BSA absorption increased with a slight blue shift of about 2 nm, indicating a static interaction between BSA and both **BPA** and **NiBPA**. The binding constants for **BPA** and **NiBPA** were calculated using Equation (1) and found to be 0.4 × 10^4^ M^−1^ and 0.82 × 10^4^ M^−1^, respectively (Table 1). These findings suggest that **NiBPA** binds to BSA more strongly than the ligand **BPA**, which is expected given the nickel center’s nature. Furthermore, the findings are in line with previous research [44,52] on related compounds.

Figure 4 shows that at low concentrations, both **BPA** and **NiBPA** have linear Stern–Volmer plots, indicating that only dynamic or static quenching is involved in these quenching processes. The *K*_SV_ values for **BPA** and **NiBPA** were found to be 0.21 × 10^4^ and 0.7 × 10^4^ M^−1^, respectively, indicating that **NiBPA** has a higher strong binding than the **BPA** ligand. The *K*_q_ value is much higher than the 2 × 10^10^ M^−1^s^−1^ maximum scatters collision quenching constant of quenchers with BSA (*K*_q_ = *K*_sv_/*τ*_0_), where K_SV_ is the Stern–Volmer constant and *τ*_0_ is the average lifetime of the BSA in the absence of the quencher (*τ*_0_ = 10 ns). This supports the idea of a static quenching mechanism. Complex **NiBPA** exhibits an upward curvature of the Stern–Volmer plot at higher concentrations, and this deviation from the linearity of the Stern–Volmer plot should occur when combined quenching (both dynamic and static) occurs, and the fluorophores can be quenched with the same quencher by both collision and complex formation [53,54] according to Equation (1):*I_0_*/*I* = (1 + *K_D_* [Q]) (1 + *K_s_*[Q]) = 1 + (*K_D_* + *K_S_*) [Q] + *K_D_ K_S_* [Q]^2^
(1)
where *K_S_* and *K_D_* are static and dynamic quenching constants, respectively. It is second order in [Q] and thus leads to upward curvy plots of *I_0_/I* versus [Q] at higher [Q]. 

The quenching constant (*K_q_*) was calculated using the Stern–Volmer and Scatchard equation and plotted as *I_0_/I vs.* [Q] to better understand the quenching process (Figure 3). In addition, Scatchard Equation (2) was used to calculate the equilibrium binding constant:log[(*I_0_*
*− I*)/*I*] = log *K*_bin_ + *n* log [Q](2)
where *K*_bin_ is the compound’s BSA binding constant and n is the number of binding sites. From the log (*I_0_ − I*)/*I vs.* log [Q] plot (Figure 5), the binding constant (*K*_bin_) and the number of binding sites (n) have been calculated (Table 1). Table 1 contains the evaluated values of the *K_q_*, *K_bin_*, and *n*. The estimated value of *n* (∼1) for **BPA** and **NiBPA** strongly supported the existence of a single binding site in BSA. The *K_q_* and *K_bin_* values for these compounds also indicated that **NiBPA** interacts with BSA more strongly than the **BPA** under investigation.

### 3.3. ct-DNA Binding Studies

Absorption titration information was depicted in Figure 6 to understand the mode of interaction of complexes with ct-DNA. Increased addition of a solution of BPA to a solution of ct-DNA causes hyperchromic for the band at 264 nm with a blue shift of 5 nm (259 nm), as shown in the absorption titration spectrum (Figure 6). The band at 278 nm, on the other hand, was hypochromic with a small blue shift of 2 nm (276 nm). The presence of more than two species in the medium was indicated by the isosbestic point in the titration curve at 274 nm. In comparison to **BPA**, the complex **NiBPA** exhibited slightly different behavior. When **NiBPA** was added to a ct-DNA solution, the band at 257 nm showed a hyperchromic shift with a minor redshift. The absorption titration studies revealed a clear interaction between the investigated complexes and ct-DNA. The hyperchromic effect, combined with a redshift, revealed the formation of a ground-state complex between ct-DNA and both **BPA** and **NiBPA**, implying the presence of a static quenching mechanism. The hydrogen bonds formed with the –OH and SH groups of **BPA** and **NiBPA** are caused by the presence of many accessible hydrogen binding sites in the DNA major and minor grooves. The binding constants for **BPA** (0.16 × 10^5^ M^−1^) and **NiBPA** (0.52 × 10^5^ M^−1^) are similar to those of well-known groove binding rather than classical intercalation, implying that **NiBPA** interacts with ct-DNA much more strongly than **BPA** under investigation.

### 3.4. Mechanistic Investigation of Protein/DNA Binding

Because many medicines with lower cellular toxicity have a slow DNA dissociation rate, several factors, such as the strength of binding and the drug’s affinity for DNA, as well as the kinetic stability of DNA-drug adducts, may contribute to antitumor effectiveness. The stopped-flow technique was used to study in-vitro protein/DNA binding to **BPA** and propose a mechanism in order to obtain such information. The kinetic traces were recorded at 265 nm within 100 s (**BPA**) and 5 s (**NiBPA**), respectively, to capture the interaction process with BSA. The absorbance of **NiBPA** increased sharply in the first 1 s of the reaction with BSA and then gradually decreased over the next 5 s. In contrast, the interaction of BPA with BSA shows a distinct pattern, with a gradual increase in absorption. In the case of ct-DNA interaction, the kinetic traces were recorded at 290 nm, and both **BPA** and **NiBPA** followed a similar pattern since kinetic traces for **BPA** and **NiBPA** were collected within 100 s, covering the entire interaction process. After a sharp decrease in the first 5 s, the absorbance gradually increased over the next 100 s. Over the concentration range of **BPA** and **NiBPA**, the study of kinetic traces performed under pseudo-first-order conditions may be fitted to two kinetic phases (Figure 7) evaluated using the following Equation (3):(3)$$A=a_{1}e^{-k_{obs1}t_{1}}+a_{2}e^{-k_{obs2}t_{2}}+A_{0}$$

The results, presented in Table 2, show that BSA binds reversibly to **BPA** with a second-order association constant of *k_1_* = 12.5 ± 1.1 M^−1^s^−1^ and dissociates from the binary complex with a first-order dissociation constant of *k_−1_* = 0.17 ± 0.0 s^−1^ in the initial phase. Complex **NiBPA** binds to BSA four times faster (*k_1_* = 53.6 ± 9.0 M^−1^s^−1^) and dissociates forty times faster (*k_−1_* = 7.3 ± 0.3 s^−1^). This means that the binding affinity of BSA to **BPA**, *K_a1_*, *k_1_/k_−1_*, of 73.5 M^−1^ is about ten times higher than that of **NiBPA** (7.3 M^−1^) and that the equilibrium dissociation constants of **BPA**, *K_d1_*, *k_−1_/k_1_*, is 13.6 × 10^−3^ M and about ten times slower than that of **NiBPA** (136 × 10^−3^ M). This means that the formed **BPA**-BSA is far more stable than **NiBPA**-BSA.

A reversible reaction with a coordination affinity of 0.69 M^−^^1^ for **NiBPA** was detected in the second reaction step, but an irreversible reaction with a *K_2_* = 0.69 M^−^^1^ with **BPA** was observed. The second phase is primarily concerned with the BSA condensation process, in addition to the electrostatic interaction. As shown in Figure 8, the irreversibility of the second reaction step with **BPA** (*k_−_*_2_ ≈ 0) suggests that the newly formed **BPA**-BSA complex is kinetically more stable in solution than **NiBPA**-BSA.

The interaction process involves at least two steps, and the stopped-flow records the two kinetic phases as a fast binding (rate constant *k_obs1_*) followed by a slow first-order isomerization process (rate constant *k_obs2_*). Both *k_obs1_* and *k_obs2_* increased linearly with **BPA** and **NiBPA** concentration (Figure 9 and Figure 10) with a slope equal to the second-order association constant, *k_on_* [M^−1^s^−1^], and an intercept equal to the first-order dissociation constant, *k_off_* [s^−1^], as demonstrated by the equation *k_obs_* = *k_off_* + *k_on_*[Comp]. Both *k_on_* and *k_off_* were used to calculate the equilibrium association constants *K_aff_* [*k_on_*/*k_off_* M^−1^] and the equilibrium dissociation constants *K_d_* [*k_off_*/*k_on_* M]. A reaction plot is shown in Figure 1, and all of the data are listed in Table 2.

The difference between the free energy, G, of BSA molecules alone in the solution and when bound together is related to the equilibrium dissociation constant, *K_d1_*. Equation (4) [55] gives the binding free energy change *G_bind_*.
(4)$$\Delta G_{bind}=RT\cdot ln(K_{d})\;$$

While *T*, *R* represents the absolute temperature and the universal gas coefficient, respectively. The Δ*G_bind_* values, calculated using Equation (4), are −10.6 and −5.1 kJ mol^−1^ for the first step binding of BSA to **BPA** and **NiBPA**, respectively. The negative sign for G indicates spontaneous binding to both **BPA** and **NiBPA**, and the greater negative value of **BPA** compared to **NiBPA** indicates that binding with **BPA** is very favorable. Both compounds interact with ct-DNA through a similar mechanism that is reversible in both reaction steps (Figure 9 and Figure 2). The initial reaction step is thought to be the complexation and production of binary complexes. **BPA** binds to ct-DNA with a second-order association constant of *k*_1_ = 27.0 ± 2.5 M^−^^1^s^−^^1^ and dissociates from the binary complex with a first-order dissociation constant of *k*_−1_ = 0.69 ± 0.1 s^−^^1^, whereas 1 binds to ct-DNA with (*k_1_* = 19.3 ± 2.3 M^−^^1^s^−^^1^) and dissociates from the binary complex with (*k_−1_* = 0.36 ± 0.1 s^−^^1^). This means that the DNA binding affinity of **1**, *K*_a1,_
*k_1_*/*k_−1_* is slightly higher (53.6 M^−^^1^) than that of **BPA** (40.0 M^−^^1^) and that the equilibrium dissociation constant of **NiBPA**, *K*_d1_, *k*_−1_/*k*_1_ is ×10^−3^ M, which is slightly slower (25.6 × 10^−3^ M). Because of the presence of a Ni(II) center, **NiBPA** has a higher affinity and a lower dissociation constant than **BPA**, indicating that the formed **1**–DNA is much more stable than **BPA**–DNA. G_bind_ values of −9.1 kJ mol^−1^ (**BPA**) and −9.7 kJ mol^−1^ (**NiBPA**) indicate that this step is spontaneous for both compounds. The data from the second reaction step, which included internal DNA rearrangement as well as electrostatic interaction and isomerization reaction, followed the same pattern as the first; a reversible reaction was observed for both **BPA** and **NiBPA**. The DNA affinity for **BPA** (*K*_a2_ = *k_2_*/*k_−2_*) is 47.5 M^−^^1^, which is slightly higher than **NiBPA** (39 M^−^^1^), and the equilibrium dissociation constant for BDA (*K*_d2_ = *k*_−2_/*k*_2_) is 2.1 × 10^−2^ M, which is slightly slower than **NiBPA** (2.6 × 10^−3^ M). *G_bind_* values for this phase are −9.4 kJ mol^−1^ (**BPA**) and −9.0 kJ mol^−1^ (**NiBPA**), indicating that this step is spontaneous for both compounds.

Equation (5) shows how to calculate the overall equilibrium dissociation constant, *K_d_,* from the individual equilibrium dissociation constants, *K_d1_* and *K_d2_*, as well as the overall association constants, *K_a_*.
(5)$$K_{d}=\frac{K_{d1}K_{d2}}{1+K_{d2}}$$

The data in Table 2 show that the DNA binding affinity of **NiBPA** (2077 M) is higher than that of **BPA** (1899). The overall reaction *G_bind_* values are −18.7 kJ mol^−1^ (**BPA**) and −18.9 (**NiBPA**) kJ mol^−1^, indicating that the reaction is spontaneous for both compounds.

### 3.5. Cytotoxic Activity

The interesting findings from BSA binding studies inspired us to use the MTT assay to test all compounds’ anticancer activity. In vitro cytotoxic activity of **BPA** and **NiBPA** were tested against human liver cancer (HepG2) and normal liver (Wi38) cell lines at various concentrations by MTT assay. The percentages of cell viability in the presence of **BPA** and **NiBPA** with different concentration levels (31.2, 62.5, 125.0, 250.0, 500.0, and 1000.0 μg/mL) are depicted in Figure 10. The cell viability was found to decrease with increasing compound concentrations for both liver cancer (HepG2) and normal (THLE2) cell lines. However, the cancer cell growth exhibited by **BPA** and **NiBPA** can be measured by the IC_50_ values and were found to be 100.0 ± 1.0 and 75.0 ± 0.5 μg/mL, respectively. From this data, one can conclude the following. Firstly, both complexes were cytotoxic active, but the activity was less than that for the normal drug (doxo, IC_50_ = 52.29 ± 1.01 μg/mL). Secondly, the Ni(II) complex **NiBPA** exhibits higher cytotoxicity compared to **BPA**. The complexation of BPA to nickel center in complex **NiBPA** may be the main factor for the better cytotoxicity of **NiBPA**. The cytotoxic selectivity of both compounds was investigated by measuring their effect on the cell viability of normal liver (Wi38) cell lines. In the case of the normal cells, the IC_50_ values suppressed by **BPA** and its Ni(II) complex **NiBPA** are 237.2 ± 1.3 and 120.0 ± 0.5 μg/mL, respectively. The IC_50_ values of the normal cells are much higher than that of the cancer cells, indicating that both compounds can be effective and selective toward cancer cells. Thus, complex **NiBPA** is the most active, indicating that nickel ion coordination was crucial for cytotoxicity and that complexation improved the anticancer activity. The positive charge in **NiBPA** due to the neutral **BPA** ligand may increase cytotoxicity. Additionally, these findings are in agreement with the affinities and binding constants of every compound that was studied for interaction with BSA.

### 3.6. Antibacterial Activity

Several studies have shown that organic compound coordination with a metallic element results in significant changes in the biological activity of both the organic ligand and the metal. As a result, **BPA** and its nickel complex **NiBPA** were tested in vitro for antibacterial activity against Gram-positive (S. aureus and S epidermidis) and Gram-negative (P. mirabilis and E. coli) bacteria. Figure 10 depicts the antibacterial function of **BPA** and **NiBPA** as the diameter of the growth-inhibition region in millimeters. Table 3 shows the minimum inhibitory concentration (MIC) and minimum bactericidal concentration (MBC) of complexes required to inhibit bacterial growth. The MBC is identified by determining the lowest concentration of antibacterial agents that reduces the viability of the initial bacterial inoculum by 99.9%. The MIC or MBC strain was chosen based on the value agreed upon on two or more occasions. The antibiotic tobramycin (TOB) was used to control the situation. The inhibition zone, as well as the MIC and MBC data, leads to the following conclusions. To begin, **NiBPA** exhibited strong antibacterial activity, and its antibacterial activity was much greater than that of BPA, but it was less than that of the standard drug (TOB). Second, **NiBPA** demonstrated greater antimicrobial activity against Gram-negative bacteria than Gram-positive bacteria. This is explained by the metal ion’s polarity and cationic charge, as **NiBPA** has a cationic property with two positive charges. Gram-negative bacteria were more sensitive to **NiBPA** than Gram-positive bacteria due to differences in cell surface characteristics; Gram-negative bacteria had a higher negative charge on their cell surface than Gram-positive bacteria [56,57]. The interaction between Gram-negative bacteria and **NiBPA** was stronger than that of Gram-positive bacteria due to the higher negative charge on the cell surface, despite the presence of an outer membrane in Gram-negative bacteria (see Figure 11).

## 4. Conclusions

New Ni(II) complexes containing bispicolylamine have been described and characterized in this study. Because almost all pharmaceuticals work by interfering with biological functions by binding to a specific protein or DNA, in vitro DNA/protein binding of **BPA** has been studied mechanistically and compared to its Ni(II) complex **NiBPA**. In comparison to **BPA**, complex **NiBPA** binds strongly to DNA/protein via a static quenching mechanism. A mechanism for the interaction was proposed using stopped-flow techniques. BSA interacts with **BPA** and **NiBPA** via two different mechanisms. **BPA**’s interaction is characterized by fast reversible followed by slow irreversible steps, whereas **NiBPA**’s interaction is characterized by two reversible steps. The binding parameters for the first step of **BPA** (*K*_a1_ = 73.5 M^−^^1^, *K*_d1_ = 13.6 × 10^−3^ M, and Δ*G*_1_ = −10.6 kJ mol^−1^) and **NiBPA** (*K*_a1_ = 7.3 M^−^^1^, *K*_d1_ = 136 × 10^−3^ M, Δ*G*_1_ = −5.1 kJ mol^−1^) and for the second step of **BPA** (*K*_2_ = 0.82 M^−^^1^) and **1** (*K*_a1_ = 3.2 M^−^^1^, *K*_d1_ = 31.2×10^−2^ M, Δ*G*_1_ = −8.3 kJ mol^−1^) were determined. Overall binding parameters for **NiBPA** (*K*_a_ = 30.3 M^−^^1^, *K*_d_ = 34.0 M, Δ*G*^0^ = −8.4 kJ mol^−1^). **BPA** and its Ni(II) complex **NiBPA** bind to DNA in a similar way, with two reversible steps: fast second-order binding followed by slow first-order isomerization. The binding parameters for the first step of **BPA** (*K*_a1_ = 40 M^−^^1^, *K*_d1_ = 25.6 × 10^−3^ M, and Δ*G*_1_ = −9.1 kJ mol^−1^) and **NiBPA** (*K*_a1_ = 53.6 M^−^^1^, *K*_d1_ = 19.0 × 10^−3^ M, Δ*G*_1_ = −9.7 kJ mol^−1^) and for the second step of **BPA** (*K*_a2_ = 47.5 M^−^^1^, *K*_d1_ = 2.1 × 10^−2^ M, and *G*_1_ = −9.4 kJ mol^−1^) and **NiBPA** (*K*_a1_ = 39 M^−^^1^, *K*_d1_ = 2.6×10^−2^ M, Δ*G*_1_ = −9.0 kJ mol^−1^) were determined. Overall binding parameters for **BPA** (*K*_a_ = 1899 M^−^^1^, *K*_d_ = 52.7 × 10^−3^ M, Δ*G*^0^ = −18.7 kJ mol^−1^) and **NiBPA** (*K*_a_ = 2077 M^−^^1^, *K*_d_ = 48.2 × 10^−3^ M, Δ*G*^0^ = −18.9 kJ mol^−1^) were also determined and showed that the relative reactivity is approximately (**BPA**)/(**NiBPA**) = 9/10. The significantly negative G values support a spontaneous binding reaction to both **BPA** and its Ni(II) complex **NiBPA**. The following points can be summarized by comparing the results of the interactions of **BPA** and **NiBPA** with DNA with those of BSA. BSA interacts with **BPA** and **NiBPA** via two different mechanisms: (i) A fast reversible followed by slow irreversible steps characterized the interaction of **BPA**, whereas two reversible steps are observed with **NiBPA**.; (ii) **BPA** and **NiBPA** interact with DNA via a similar mechanism via two reversible steps; (iii) Complex **NiBPA** was found to have much higher affinity to DNA (2077 M) than BSA (30.3 M), despite the fact that BSA interacts with this, which is explained by the fact that the fast association of **NiBPA** with BSA is followed by a fast dissociation reaction; (iv) **BPA** formed a more stable BSA complex than its Ni(II) complex. This is also understandable because they both bind via different mechanisms, with the second binding step for **BPA** being irreversible and the second binding step for **NiBPA** being reversible; (v) Interestingly, when **BPA** and **NiBPA** bind to both DNA, the Ni(II) center was found to affect the rate but not the mechanism. In the case of **BSA** interaction with **BPA** and its Ni(II) complex, the Ni(II) center was found to change both the mechanism and the rate. Additionally, **NiBPA** exhibited significant cytotoxicity against the human liver cancer HepG2 cell as well as antibacterial activity, which is consistent with the binding constants but not the kinetic stability. This shows that in our situation, cytotoxic and antibacterial activities are significantly more influenced by binding constants than by kinetic stability. Due to its selectivity and cytotoxic activity, the complex **NiBPA** is anticipated to be used in medicine.

## Data Availability

Not applicable.

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
