# Peer review of "Synthesis, Biophysical Interaction of DNA/BSA, Equilibrium and Stopped-Flow Kinetic Studies, and Biological Evaluation of bis(2-Picolyl)amine-Based Nickel(II) Complex"

_biomimetics, 2022, doi:10.3390/biomimetics7040172_

Round 1

Reviewer 1 Report

The authors reported the synthesis of novel nickel complex and their binding affinity (also mechanism) to DNA/BSA. This is an interesting study, however, the identification of synthetic compounds is inadequate. In addition, there are many lacks in the evaluation of the biophysical interaction, and the authors should correct the spectral data and add explanations. Based on these points, this paper needs to be drastically rewritten in its composition and is not worthy of acceptance in biomimetics at this stage.

Some key points are noted below.

1.      The molecular formula of bis(2-picolyl)amine is C12H13N3, and the complex with NiCl2/H2O would be C12H15Cl2N3NiO, but elemental analysis shows C13H17Cl2N3NiO. The authors should explain this.

2.      The left (IR) and right (UV) sides of Figure 1 should be replaced so that they are in the same order as described in the text (since the text describes UV results first and the IR results later). The left and right spectra of Figure 6 should also be replaced (same reason).

3.      The UV of the titration of NiBPA to CT-DNA shows an increase in the baseline, suggesting that it is in suspension (as the UV change may not be due to complex formation). The authors should check again.

4.      The authors should also explain the absorption around 700 nm in the UV spectrum.

5.      Why is there an increase in absorbance for both BPA and NiBPA in normal UV (Figure 4), but a decrease in NiBPA in stopped flow (Figure 6)?

6.      What does A350nm in Figure 6 mean? At what wavelength are the authors tracking absorption?

Author Response

Response to the reviewer’s comments (biomimetics-1942923)

Dear Prof. Dolly Liu,

Thank you for your letter of 04-Oct-2022 and comments from the reviewers. We have revised the manuscript and have responded (in blue) and highlighted the text (in yellow) to the comments as follows:

------------------------------------------------------------------------------------------------

Referee (1)

Comment: The authors reported the synthesis of novel nickel complex and their binding affinity (also mechanism) to DNA/BSA. This is an interesting study.

Response: We appreciate very much the referee’s comments.

Comment: however, the identification of synthetic compounds is inadequate. In addition, there are many lacks in the evaluation of the biophysical interaction, and the authors should correct the spectral data and add explanations. Based on these points, this paper needs to be drastically rewritten in its composition and is not worthy of acceptance in biomimetics at this stage. Some key points are noted below.

Comment 1. The molecular formula of bis(2-picolyl)amine is C12H13N3, and the complex with NiCl2/H2O would be C12H15Cl2N3NiO, but elemental analysis shows C13H17Cl2N3NiO. The authors should explain this.

Response: Yes, we are very sorry, there was a mistake and it was corrected in the revised manuscript.

Comment 2. The left (IR) and right (UV) sides of Figure 1 should be replaced so that they are in the same order as described in the text (since the text describes UV results first and the IR results later). The left and right spectra of Figure 6 should also be replaced (same reason).

Response: Yes, we agree with the reviewer and the figures were modified as suggested by the reviewer, thank you!

Comment 3. The UV of the titration of NiBPA to CT-DNA shows an increase in the baseline, suggesting that it is in suspension (as the UV change may not be due to complex formation). The authors should check again.

Response: We have checked the spectra again and extended the wavelength to 600 nm and have redrawn again. We have also recorded Fig 6 to be in the order of the discussion in the text.  

Comment 4. The authors should also explain the absorption around 700 nm in the UV spectrum.

Response: The band has been explained and added to the text with references.  

Comment 5. Why is there an increase in absorbance for both BPA and NiBPA in normal UV (Figure 4), but a decrease in NiBPA in stopped flow (Figure 7)?

Response: In the normal UV, we are detecting the overall steps as it takes longer time to be measured, whereas, for stopped-flow we are detecting every step individually and the only one confusing is NiBPA with BSA. In this case we can investigate only two reaction steps.  

Comment 6. What does A350 nm in Figure 6 mean? At what wavelength are the authors tracking absorption?

Response:  A350 means that we are tracking absorption at 350 nm. This is the answer for the second part of the comment.  

Referee (2)

Comment: This paper is well documented about the biological evaluation of bis(2-picolyl)amine-based Nickel(II) complex.

Response: We appreciate very much the referee’s comments.

Comment: However, bis(2-picolyl)amine-based Nickel(II) complexes are well-known compounds, but in this paper literature survey about the biological properties of these compounds is not surveyed. Hence, additional information is requested:

Response: We agree with the reviewer and some sentences and references were added to the text

We do believe that the revised manuscript has now been revised satisfactorily responding to the referees’ comments.

Reviewer 2 Report

This paper is well documentated about the biological evaluation of bis(2-picolyl)amine-based Nickel(II) complex. However, bis(2-picolyl)amine-based Nickel(II) complexes are well-known compounds, but in this paper literature survey about the biological properties of these compounds is not surveyed. Hence, additional information is requested.

Author Response

(The authors gave the same response as above.)

Round 2

Reviewer 1 Report

About Comment 1: The results of elemental analysis show a deviation of more than 1% between the theoretical value and the experimental value. This does not provide structural proof of this compound, and therefore, the authors should re-measure, or measure by HRMS, or consider the presence of solvents, etc.

About Comment 5: In Figure 7, why do the authors trace absorption changes at 350 nm? Figure 4 would not have shown absorption changes in those regions. Also, the authors did not show which absorption wavelengths were traced in NiBPA-BSA. Thus, these rate analysis data do not make sense because it is unclear which wavelength change, i.e., what kind of structural change, is being studied.

Author Response

Response to the reviewer’s comments (biomimetics-1942923 Round 2)

Dear Prof. Dolly Liu,

Thank you for your letter on 13-Oct-2022 and the reviewer comments. We have revised the manuscript and have responded (in blue) and highlighted in text (in green) to the comments as follow:

------------------------------------------------------------------------------------------------

Referee (1)

About Comment 1: The results of elemental analysis show a deviation of more than 1% between the theoretical value and the experimental value. This does not provide structural proof of this compound, and therefore, the authors should re-measure, or measure by HRMS, or consider the presence of solvents, etc.

Response: The EA has been re-measured for the same sample used in the experiments after extra drying and was corrected in the text.

About Comment 5: In Figure 7, why do the authors trace absorption changes at 350 nm? Figure 4 would not have shown absorption changes in those regions. Also, the authors did not show which absorption wavelengths were traced in NiBPA-BSA. Thus, these rate analysis data do not make sense because it is unclear which wavelength change, i.e., what kind of structural change, is being studied.

Response: Thank you for noting this! Off course 350 nm is not the appropriate wavelength, this was a mistake and we recorded the kinetic traces of both compounds with BSA at 265 nm which showed the maximum absorbance change where the peak of BSA is coming and originated from n-π* transitions of amino acids such as phenylalanine, tyrosine, and tryptophan of BSA. The figure has been corrected.

We do believe that the revised manuscript has now been revised satisfactorily responding to the referees’ comments.